# Data-Driven and Model-Driven Joint Detection Algorithm for Faster-Than-Nyquist Signaling in Multipath Channels

**DOI:** 10.3390/s22010257

**Published:** 2021-12-30

**Authors:** Xiuqi Deng, Xin Bian, Mingqi Li

**Affiliations:** 1Shanghai Advanced Research Institute, Chinese Academy of Sciences, Shanghai 201210, China; dengxiuqi2019@sari.ac.cn (X.D.); bianx@sari.ac.cn (X.B.); 2University of Chinese Academy of Sciences, Beijing 100049, China

**Keywords:** FTN signaling, deepearning, detection, multipath, data-driven and model-driven combination, neural network adaptability

## Abstract

In recent years, Faster-than-Nyquist (FTN) transmission has been regarded as one of the key technologies for future 6G due to its advantages in high spectrum efficiency. However, as a price to improve the spectrum efficiency, the FTN system introduces inter-symbol interference (ISI) at the transmitting end, whicheads to a serious deterioration in the performance of traditional receiving algorithms under high compression rates and harsh channel environments. The data-driven detection algorithm has performance advantages for the detection of high compression rate FTN signaling, but the current related work is mainly focused on the application in the Additive White Gaussian Noise (AWGN) channel. In this article, for FTN signaling in multipath channels, a data and model-driven joint detection algorithm, i.e., DMD-JD algorithm is proposed. This algorithm first uses the traditional MMSE or ZFinear equalizer to complete the channel equalization, and then processes the serious ISI introduced by FTN through the deepearning network based on CNN or LSTM, thereby effectively avoiding the problem of insufficient generalization of the deepearning algorithm in different channel scenarios. The simulation results show that in multipath channels, the performance of the proposed DMD-JD algorithm is better than that of purely model-based or data-driven algorithms; in addition, the deepearning network trained based on a single channel model can be well adapted to FTN signal detection under other channel models, thereby improving the engineering practicability of the FTN signal detection algorithm based on deepearning.

## 1. Introduction

With the development of mobile multimedia services such as high-definition video and XR, mobile communication systems have put forward higher and higher requirements for transmission rates. In the current situation of increasingly tight spectrum resources, improving the spectrum efficiency of the transmission system has increasingly become a necessary way to improve the throughput of the communication system [1]. As a non-orthogonal transmission technology that can achieve higher spectral efficiency, FTN has attracted widespread attention in academia and industry worldwide.

FTN realizes the compression of the transmitted signal in the time domain and the frequency domain by introducing a certain amount of ISI at the transmitting end in advance [2,3,4,5], so as to obtain higher spectral efficiency than traditional orthogonal transmission technology. As a non-orthogonal transmission technology, FTN breaks the traditional Nyquist criterion. In the case of artificially introducing inevitable ISI, Mazo’s research shows that the data rate can be increased by about 25% without performanceoss. Although the existence of Mazo Limit enables FTN transmission to improve the spectrum efficiency of the system while ensuring the reliability of the transmission, because the FTN system will inevitably introduce ISI, it is necessary for the receiving end to use detection techniques to eliminate ISI. However, compared to the orthogonal system, the receiver only needs to go through simple matched filtering to achieve the best detection symbol-by-symbol, a non-orthogonal FTN system requires more complex receiver processing algorithms to eliminate interference caused by ISI in order to obtain real gains. Therefore, it can be considered that designing a detection algorithm with good performance and acceptable computational complexity is the key to FTN research. In recent years, research on FTN has mainly focused on model-driven detection algorithms, but for FTN signaling with higher compression rates, whether it isinear detection algorithms based on MMSE or ZF, or non-linear detection algorithms such as MAP or BCJR, the performance is not ideal, and theatter’s implementation complexity is even very high. In order to overcome the above-mentioned problems, the FTN signal detection algorithm based on data-driven has gradually attracted the attention of scholars. Preliminary research results show that under the condition of a high compression rate, the performance of the FTN signal detection algorithm based on deepearning is better than the traditional model-driven algorithm. However, due to the huge differences between training data and test data in time-varying multipath channels in terms of channel characteristics such as multipath delay expansion and moving speed, the FTN signal detection algorithm based on deepearning has a serious problem of insufficient generalization. Therefore, the current related research mainly focuses on FTN signal detection in the AWGN channel or the microwave communication environment that can be similar to AWGN.

In this article, we propose a DMD-JD detection algorithm to solve the problem of FTN signal detection in time-varying multipath channels. The model-driven part adoptsinear equalization technology to eliminate the influence of channel multipath fading to improve the generalization ability of the model. The data-driven part adopts the Deep-Bi-LSTM deepearning network model based on Long Short-Term Memory (LSTM) and uses the two-way network structure to obtain the ISI information caused by the adjacent symbols before and after the symbol of each FTN signaling to eliminate the influence of ISI caused by FTN signaling compression. The channel equalization adopts theinear equalization algorithm, and the deepearning network can complete the training offline. Therefore, the overall implementation complexity of the proposed DMD-JD detection algorithm is stillower than that of the traditional nonlinear detection algorithm, which improves the engineering practicability of the algorithm.

In general, the main contributions of this article include:Combining the data-driven and the communication model-driven prior knowledge, through the introduction of the channel equalization module, it is not only more effective than the pure data-driven FTN detection algorithm. Additionally, to a certain extent, the problem caused by the data mismatch under the multipath channel is alleviated, so that the data sets under different channel states are adaptable, thereby improving the robustness of the proposed algorithm.The LSTM network is applied to FTN signal detection in multipath channels, and a Deep-Bi-LSTM structure more suitable for multipath channel FTN detection is obtained by adjusting appropriate parameters, loss function, and network structure. Compared with traditional algorithms, it not only hasower complexity but also can obtain better ISI suppression performance for FTN signaling.Through aarge number of simulations, the performance advantages of the proposed algorithm compared with the existing pure model-driven and pure data-driven algorithms are verified. The effects of different channel equalization algorithms and different deepearning networks on the performance of the proposed algorithm are evaluated, and the effectiveness of the proposed scheme under QPSK modulation is also verified. In addition, the network obtained by data training under a fixed channel model is used to test the data under other channel models to verify the adaptability of the proposed DMD-JD detection algorithm to actual time-varying multipath channels.

## 2. Related Work

The concept of FTN was first introduced by James Mazo of Bell Labs in his 1975 research results [6], where he found that for a PAM(Pulse Amplitude Modulation) transmission system that uses SINC Pulses as a shaping filter, when the symbol interval isower than the Nyquist criterion and is transmitted in the AWGN channel, the minimum Euclidean distance between signals will not decrease in the symbol interval τT within the range of 0.802 ≤τ≤ 1. This means that the system can increase the spectrum efficiency by 25% while ensuring that the bit error rate performance is not affected. FTN re-examined the relationship between signal bandwidth, symbol rate, and spectrum efficiency, but it was not taken seriously at that time due to theimitations of hardware conditions. After a period of silence, literature [7] observed that the FTN system formed by root raised cosine (RRC) also has Mazo-limited properties. Subsequently, the team from Sweden completed a series of groundbreaking studies on the development of FTN. They have successively confirmed that the Mazoimit is universal in the frequency domain [8], high-order modulation [9], multiple-antenna Multiple-Input and Multiple-Output (MIMO) [10], carrier [11], and other systems. In addition, it was proven that when the time domain compression factor τ is set appropriately, the FTN system can achieve a higher channel capacity than the general orthogonal modulation system [12]. The research on FTN has gradually heated up in recent years, and the design of the FTN detection algorithm is the top priority of FTN research.

Existing common model-driven FTN signal detection technologies mainly include methods based on maximumikelihood sequence estimation (MLSE) or maximum a posteriori (MAP). As part of the ISI information is known, and FTN signaling can be equivalently regarded as trellis coding, most of them are realized by the Viterbi algorithm (VA) [13] or BCJR [14] algorithm. The complexity of these optimal methods increases exponentially with the number of ISI taps and modulation order considered. Therefore, there are many efforts to find sub-optimal implementations. For example, the work in [15] introduces the successful symbol-by-symbol with go-back-K sequence estimation (SSSgbKSE) detection technology into the polar codes of FTN signaling, which is closer to the BCJR algorithm while reducing the complexity. The authors of [16] proposed an FPGA-based sliding-window max-log MAP algorithm that can be directly applied to FTN transmission systems to mitigate ISI. Theiterature [17] proposed a convex quadratic relax-and-quantize sequence estimation (CQRAQSE) algorithm suitable forow-rate FTN transmission based on convex relaxation. Although suboptimal implementation algorithms based on MLSE or MAP can achieve good detection accuracy, the implementation complexity is always exponential. Compared with the exponential complexity algorithm with higher complexity, the frequency domain equalization based oninear complexity hasower complexity, but the performance is relatively poor. Compared with the exponential complexity algorithm with higher complexity, the frequency domain equalization based oninear complexity hasower complexity, but the performance is relatively poor. The article [18] proposed a cyclic block transmission scheme, namely CB-FTN, which uses cyclic convolution instead ofinear convolution for pulse shaping, andow-complexity frequency domain equalization can be used at the receiver to compensate for channel damage and Inherent ISI. In reference to the CB-FTN scheme, the authors of [19] proposed a transceiver equivalent implementation scheme based on DFT (DBT-FTN), which can effectively reduce the complexity of base-band signal processing at both ends of the transceiver. The works in [20,21] extended the DBT-FTN transmission scheme from single-carrier to multicarrier, forming a multicarrier DBT-FTN (DBT-MC-FTN) scheme. Literature [22] proposed a joint channel estimation and precoding (JCEP) algorithm for data detection of FTN signaling on frequency selective fading channels.

With the development of AI technology, the problem of data-driven communication signal detection has received increasing attention. On the one hand, various deepearning methods have recently been used to solve the problem of the physicalayer of wireless communication, and valuable results have been obtained [23]. In addition, the signal training process of LSTM has passed the basic principle of signal processing, and the LSTM expansion model has been formally explained in the framework of approximating the IIR system with the FIR model, and it has been proven to be suitable for sufficient conditions forearning signal sequences [24]. Yet, on the other hand, as far as our research has uncovered, no or very few scholars use data-driven methods to solve FTN detection algorithms in multipath channels. The existing data-driven related work mainly stays in AWGN channel or special scene channel (similar to AWGN optical communication channel and underwater communication channel) [25,26,27]. Preliminary research results show that for FTN signaling with a higher compression rate, the detection algorithm based on deepearning can obtain better performance than the MAP algorithm [28]. Compared with the FTN research in the AWGN channel, the research of the FTN signaling receiving algorithm based on the multipath fading channel has more practical value. However, FTN signaling transmission in multipath channels will also bring major challenges to existing receiver algorithms. This is because the inherent ISI of FTN signaling and the non-linear interference caused by the multipath channel are aliased with each other, which makes it difficult for traditional detection algorithms to recover the correct transmitted signal. In addition, most of the existing data-driven FTN detection algorithms use traditional neural networks such as DNN and CNN as the basic structure, and there are relatively few structures that use RNN as a model. On the one hand, RNN networks are widely used in the field of signal processing by virtue of their excellent sequence processing capabilities. On the other hand, our simulation results also show that the RNN network structure is better than the traditional neural network structure for feature extraction of FTN signal interference. This article will propose a new idea for FTN detection with great potential. Compared with traditional methods with perfect mathematical arguments, neural network detection algorithms help to achieve online real-time detection by characterizing training samples offline in advance. With this method, improvements with better robustness andower complexity can be obtained. In general, as the spectrum resources areimited and the computing power is increasing day by day, FTN and data-driven signal processing algorithms are promising development directions and are expected to play an important role in the future communication physicalayer transmission technology.

## 3. FTN System Model Design

### Transmission System Model Based on DMD-JD FTN Signal Detection Algorithm

Figure 1 shows the principle block diagram of the communication system using the DMD-JD FTN detection algorithm based on the data-driven Deep-Bi-LSTM structure in the multipath channels. Among them, the ISI actively introduced by FTN at the transmitting end is obtained by using τT as the sampling interval in the up-samplingink of the shaping filter. It can be seen that the overall process of FTN signaling generation is roughly the same as the signal generation process of a traditional communication system. In the data preprocessing stage, targeted preprocessing will be done due to different modulation methods. This allows the neural network to extract effective features from the signal data to eliminate the introduced ISI and its nonlinear aliasing after passing through the channel. For example, taking QPSK (4QAM) modulation as an example, it is assumed that the transmitting end sends a transmission bit sequence with aength of *N*, which corresponds to a receiving end symbol sequence with aength of N/2 after down-sampling. At the receiving end, the received symbol sequence is divided according to the real part and the imaginary part, and the data set is constructed according to the principle of one-to-one pairing with the transmitted symbols.

What needs to be emphasized is the role of the channel equalization module. In the purely data-driven situation, due to the time-varying characteristics of the multipath channel, it will be difficult to match the training data and the test data, which willead to a sharp deterioration in the performance of the test results and it is difficult to achieve the desired performance. The increase of the channel equalization module can alleviate the model training problem caused by the data mismatch. In addition, the introduction of the equalization module can also achieve better generalization performance in various multipath channel switching scenarios, thereby improving the adaptability and robustness of the system.

Consider the binary phase-shift keying (BPSK) modulated FTN signaling system, assuming that the information symbol is denoted as a(d). Where *d* and *D* respectively represent the symbol index and the number of symbols in each information symbol. After FTN shaping and filtering, the modulated base-band continuous signal can be expressed as
(1)$$s\left(t\right)=\sum_{d=0}^{D-1}a\left(d\right)p\left(t-d\tau T\right),0<\tau <1$$p(t) is a pulse-shaped filter with T as the orthogonal shift interval, theength of the filter is L, and τ is the time-domain compression factor. Sampling with ΔT as the interval, we can obtain
(2)$$s\left(n\right)=\sum_{d=0}^{D-1}a\left(d\right)p\left(n-dN_{f}\right),0\leq n\leq \left(D-1\right)\times N_{f}+L-1$$
where $\tau T=N_{f}\Delta T$, ${\tau \;=\;N}_{f}/N_{s}$. $N_{f}$ is the pulse-shaped filter shift interval, and $N_{s}$ is the up-sampling rate of the pulse-shaped filter, which satisfies as
(3)$$\sum_{0}^{L-1}p\left(n\right)p^{*}\left(n-dN_{s}\right)=\left\{\begin{array}{c} 1,d=0\\ 0,d\neq 0 \end{array}\right.$$

After going through the multipath channel $h\left(t\right)$, our signal can be expressed as
(4)$$y\left(n\right)=s\left(n\right)⊗h\left(n\right)+w\left(n\right)$$
where h(n) is the channel impulse response, w(n) is the white noise generated by the AWGN channel with variance ${\;\sigma }^{2}$, and ⊗ represents the cyclic convolution operation. Performing DFT on y(n) can transform the time domain convolution into the form of the frequency-domain product. After passing through the multipath channel, perform IDFT on the received signal to get the received signal at the receiving end $y\left(d\right)$.
(5)$$y\left(d\right)=\sum_{d^{\prime }=0}^{D-1}a\left(d^{\prime }\right)g\left(d-d^{\prime }\right)+\int_{-\infty }^{+\infty }w\left(n\right)p^{*}\left(n-dN_{f}\right)dn$$

The second term in Equation (Equation 5) is the colored noise after Gaussian white noise and matched filter convolution. $g(d-d^{\prime })$ is the channel equalization, the FTN signal passes through the multipath channel through the shaping filter and then convolves with the matched filter after the FTN system actively introduces the ISI coefficients and the convolution sum of the channel. That is, the ISI tap coefficients are aliased in the nonlinear interference generated by the fading channel. $g(d-d^{\prime })$ can be expanded to
(6)$$g\left(d-d^{\prime }\right)=\int_{-\infty }^{+\infty }p\left(d^{\prime }-d\tau T\right)p^{*}\left(d^{\prime }-dN_{f}\right)\cdot \sum_{l=-L_{B}}^{L_{F}}h_{l}\left[dN_{f}-\left(l+d^{\prime }\right)N_{f}\right]$$
where $L_{F}$ and ${\;L}_{B}$ represent the number of forward and backward multipaths, respectively, and $L_{D}=\;L_{F}+\;L_{B}\;+\;1$ represents the delay spread of the frequency selective fading channel.${\;h}_{l}$ represents the influence of the channel gain of the *l*th channel on the current received signal. That is, after the FTN signal passes through the multipath channel, the non-linear noise interference that it receives can be approximately regarded as the non-linear ISI interference and passes through a non-linear multipath channel. The aliasing of two non-linear interferences will pose a major challenge to the existing receiver algorithms. In particular, when $L_{F}\;={\;L}_{B}=\;0$, the multipath channel is equivalent to the AWGN channel, and the interference will degenerate into only the ISI introduced by the convolution of the shaping filter and the matched filter.

The noise elimination in the system is divided into two modules. First, the ideal channel equalization is used to offset most of the multipath channel interference. The filtered signal is then subjected to data-driven FTN signal detection to eliminate the ISI introduced at the transmitting end and the nonlinear aliasing after the ISI passes through the multipath channel. The description of the signal detection module will be described in detail in the next section, and the specific simulation results will be explained in detail in Section 5.2.

## 4. FTN Detection Scheme in Multipath Channels Based on Neural Network

The shaping filter withength L-length is expressed as $p={\left[p_{1},p_{2},\cdots ,p_{L}\right]}^{T}$, a is expressed as N-length independent equal probability According to the content of the previous section, the following discrete form of the FTN signal vector model can be obtained
(7)s=Pa
where s represents the transmission symbol sequence transmitted by the transmitting end after shaping by a shaping filter with a period of τT, $P=\left[p_{1},p_{2},\cdots ,p_{N}\right]\in R^{L\times N}$ represents the pulse-shaping matrix. After the FTN signal passes through the multipath channel, the received symbol at the receiving end can be equivalently modeled as a process through the nonlinear channel as follows
(8)$$r=P^{*}\left(\mathrm{Hs}+w\right)=Ga+\eta$$
where w is the Gaussian white noise signal, η is the colored noise generated after matched filtering, H is the channel response matrix of the multipath channel, and it is also a cyclic matrix. $G=\;P^{*}\mathrm{HP}$ is expressed as the non-linear aliasing of ISI introduced by FTN and multipath channel, the inter-symbol interference matrices introduced by FTN are represented as ${\;P}^{*}$ and P, respectively.

In order to recover the originally transmitted signal from the non-linear interference aliased received signal, two different deepearning models, CNN and RNN, will be used to detect FTN signaling.

### 4.1. Detection via CNN

In the CNN model, we treat FTN detection as a classification problem, select the signal that has not been matched and filtered at the receiving end as the sample set, and theabel set is the symbol sequence of the transmitted signal.The specific structure of using a convolutional neural network as a classification model is shown in Figure 2. Both convolutionalayers use Relu as a non-linear activation function. Finally, after passing through the fully connectedayer, the output probability distribution is obtained from the activation function SoftMax, where the symbol corresponding to the maximum probability is the estimated signal $\hat{a}$.

The rationality of using CNN as the detection algorithm can be explained as follows. In FTN, the influence of the value of the symbol before and after the symbol on the judgment of the current symbol must be considered. From this perspective, the process of the filter sliding on the input data in the CNN model to extractocal features can be regarded as a series of related operations of a small-sized matched filter at different time positions, which measures the magnitude of interference generated by symbols at different moments and uses this as a feature to help subsequent classification judgments.

For high-order modulation, since the information bits are stored in the real and imaginary parts of the signal, the signal at the receiving end is separated according to the real and imaginary parts and then integrated into the corresponding transmission symbol sequence. Specifically, IQ signals can be connected in parallel and the real and imaginary parts can be input into the networkine byine, which is equivalent to processing single-channel two-dimensional images (such as grayscale images). A more reasonable method is to put the real and imaginary signals on different channels, that is, treat the real and imaginary data as different images. This process can be analogous to the three-channel processing of RGB in digital image processing. Furthermore, as the noise interference of FTN signaling in the multipath channel is independent and identically distributed, the correlation between the real part and the imaginary part of the noise is faress clear than the correlation between the adjacent real part or imaginary part signals. Therefore, it can consider using the real part and the imaginary part to be connected to different channels respectively, and the corresponding feature information is extracted, and then the classified results are merged and outputted.

The advantage of the symbol-by-symbol detection of the CNN model is that the model is simple, but the sample signal is not clear enough to express ISI. Therefore, on this basis, we try to use the idea of discrete signal sequence detection to deal with the problem of strong correlation in the time dimension of the FTN—RNN network model.

RNN allows information to persist, that is, the current output depends on all previous inputs. From a computational point of view, as the current state depends on previous calculations, and the network performs the same task for each element of the input sequence. In the next section, the Deep-Bi-LSTM network based on RNN will be introduced in detail, and the effectiveness of FTN detection in multipath channels will be analyzed.

### 4.2. Proposed Deep-Bi-LSTM Architecture for FTN Signal Detection

In the previous section, FTN signal detection is regarded as a classification problem. Consider using the CNN network to do different processing methods for different received signal forms. In this section, we treat FTN signal detection as a regression problem and explain the proposed Deep-Bi-LSTM network model in detail.

#### 4.2.1. Use RNN-Based LSTM Unit for FTN Detection

Existing research has proven that the RNN-based sequence detection architecture has proven effective in many fields. This is because the RNN network can map the effects of all previous inputs to the current output through feedback connections, which makes the RNN network ‘memorable’ and causal. Aiming at the problem of FTN signal detection, these characteristics of the RNN network are suitable for solving this problem of strong correlation in the time dimension. The proposed network architecture is shown in Figure 3.

The input of the model is the preprocessed N-length received symbol sequence. After extracting features through two identical fully connected hiddenayers using Bi-LSTM, the fully connectedayer with tanh as the activation function obtains an output in the range (−1, +1). In addition, batch normalization is added after each hiddenayer operation to correct the data distribution to prevent gradient vanishing problems that are difficult toearn effectively. Unlike traditional two-classification problems that mostly use cross-entropy as theoss function, we found in the actual test that using the l2 norm as theoss function can achieve relatively better performance in this problem. The possible reason is that the activation function in the fully connected network chooses to use tanh instead of sigmoid.

The sequence detection model directly outputs the estimated symbol at each moment as ${\hat{a}}_{k}$, and the correspondingoss function is
(9)$$L\left(a,\hat{a}\right)=\frac{1}{N}\sum_{k=1}^{N}{\left|a_{k}-{\hat{a}}_{k}\right|}^{2}$$

In the process of using traditional RNN, theoretically, it can handle arbitrarilyong sequences. However, in actual application, it often encounters aong-term dependencies problem. Generally, aong short-term memory (LSTM) unit [29] is used to alleviate this problem.

#### 4.2.2. Reasons for Using Bi-LSTM Structure

In FTN signal detection, it is necessary to consider that the shaping filter is axisymmetric, which means that for the current time k, the interference caused by the symbols at k−1 and k+1 is of the same magnitude. It is obviously not enough to only use the ISI before time k as the interference feature of the current signal. The method to solve this problem is to introduce the Bi-directional LSTM model. As shown in the yellow part of Figure 3, the bidirectional cyclic neural network model includes a cyclic neural network with a front-to-back processing sequence and a cyclic neural network with a back-to-front processing sequence, which are used to transmit the forward state $\vec{h_{t}}$ and the backward state $\overset{\leftarrow }{h_{t}}$.
(10)$$\vec{h_{t}}\left[n\right]=\mathrm{sigmoid}\left(W_{x,cr}^{\prime }\vec{x}\left[n\right]+W_{c,cr}^{\prime }\vec{c}\left[n\right]+W_{h,cr}^{\prime }{\vec{h}}_{t}\left[n+1\right]+{\vec{b}}_{cr}^{\prime }\right)⊙\tanh\left(\vec{c}\left[n\right]\right)$$
(11)$$\overset{\leftarrow }{h_{t}}\left[n\right]=\mathrm{sigmoid}\left(W_{x,cr}^{\prime }\vec{x}\left[n\right]+W_{c,cr}^{\prime }\vec{c}\left[n\right]+W_{h,cr}^{\prime }{\overset{\leftarrow }{h}}_{t}\left[n+1\right]+{\vec{b}}_{cr}^{\prime }\right)⊙\tanh\left(\vec{c}\left[n\right]\right)$$
where $\vec{x}\left[n\right]$ is the model input, $\vec{c}\left[n\right]$ is the information storage unit in the LSTM, and *W* and $\vec{b}$ are the weights and offsets.

Then, the feature information from the pre-order traversal and the post-order traversal is feature merged. Fusion strategies generally include addition, multiplication, averaging, or simple concatenation, etc. The feature fusion method selected here is the default concatenation, where ⊕ represents the integration operation.
(12)$$h_{t}=\left[{\vec{h}}_{t}⊕{\overset{\leftarrow }{h}}_{t}\right]$$

#### 4.2.3. Reasons for Using Deep-LSTM Structure

Like Deep Neural Networks (DNN) with a deep architecture, Deep-LSTM has been successfully used in fields such as speech recognition text generation, machine translation, speech recognition, generated image description, and video tagging. In fact, any single LSTM neural network is already a deep architecture, because it can be regarded as a feed-forward neural network unfolded in time, in which eachayer shares the same model parameters. Compared with traditional standard LSTM networks, Deep-LSTM networks provide another advantage. They can make better use of parameters by distributing parameters in multipleayers in space. For example, if the memory size of the standard model is increased by 2 times, there can be 4 layers with roughly the same number of parameters, whicheads to more non-linear operations for the input of each time step. The simulation in Section 5.2 also proves that introducing a Bi-LSTM network with a suitable depth can improve the performance of the network for FTN signal detection, especially in multipath channel scenarios.

All in all, the interference of FTN signaling mainly comes from the influence of adjacent symbols on the current symbol, which has something in common with the processing architecture of Deep-Bi-LSTM.

### 4.3. Dataset

The data sets are all FTN signaling in the multipath channels obtained through the same communication system. The data sets can be divided into training data sets and test data sets according to their functions. Theength of each FTN signaling is N, and this signaling represents independent data obtained in software simulation. In the data set, we use the transmitted symbol as theabel, and the corresponding receiving end receives the FTN signaling as the sample. The real part of the signal is regarded as the sample in BPSK modulation, and the real and imaginary parts of the signal will be respectively regarded as the sample in QPSK modulation. In the training and testing phase of the network, the corresponding “label–sample” pairs are used as the data set of the Deep-Bi-LSTM network.

### 4.4. Complexity Analysis

*M* represents the modulation order, and $L_{ISI}$ represents theength of the FTN signaling affected by the ISI considered by the algorithm. It should be noted here that the ideal theoretical value of $L_{ISI}$ should be asong as the sequence, that is, ISI interference comes from the entire sequence. However, existing algorithms are generally set at a fixed value to reduce the difficulty of implementation, which reduces performance to a certain extent.

The existing traditional algorithms includeinear-level complexity algorithms and exponential-level complexity algorithms. Theower complexity is the frequency domain equalization algorithm, whose complexity is $O\left(N\right)$, while the complexity of the MAP algorithm is $O\left(L_{ISI}{NM}^{2L_{ISI}+1}\right)$. For data-driven algorithms, the training process is completed offline in advance and does not occupy computing resources, so only the complexity of online operations needs to be considered. Since the CNN model used is determined to be twice the dimensionality of the input data when processing two-dimensional input, the complexity measurementevel of the convolutionalayer operation is the same for the same dimensional input, which is both $O\left(N\right)$. For the DMD-JD algorithm, the computational complexity of the online training stage is $O\left(N^{2}\right)$, but in the real-time detection stage, as there is only simple forward reasoning, the computational complexity will be reduced to $O\left(N\right)$. Although the calculation amount of Deep-Bi-LSTM is four times that of LSTM, it still has ainear relationship with the sequenceength.

What needs to be emphasized here is that the complexity of data-driven algorithms does notie in their operations, but transfers the complexity to model adaptability. Specifically, due to the time-varying characteristics of the channel, the data characteristics of different channels, different moving speeds, and even different signal-to-noise ratios are different, and targeted model training needs to be performed according to specific application scenarios, which willead to higher complexity. For the problem of generalization performance, we present targeted work in Section 5.2.3 to make up for this defect.

## 5. Experimental Evaluation

### 5.1. Experimental Setup

In this section, the system performance of the FTN signal detection scheme will be evaluated in different channel simulation scenarios. Under the 64-bit Windows 10 operating system, the simulation was performed on the Intel (R) Core (TM) i7-6700K @ 4.00 GHz CPU, Nvidia GeForce GTX 970 (4 GB / Nvidia) on Matlab2018B and PyCharm platforms. Among them, the training and test data sets are randomly generated on the Matlab simulation platform through different channels. The proposed neural network is implemented by Keras, written in Python, and uses TensorFlow as the backend to perform tensor operations. The main parameters of the FTN signal are shown in Table 1, the main simulation parameters of deepearning are shown in Table 2, and the main parameters of the system in multipath channel analysis are shown in Table 3 [30].

Unless otherwise specified, the mobile rate of the simulated multipath channel is the result of aower mobile rate by default, that is, TU-6 is 30 km/h, PB is 3 km/h, and VA is 60 km/h. The hyperparameters Dropout rate and Num unit are updated based on the validation set (the same way as the training set test set is generated). Specifically, in the training phase of the model, after training for a certain epoch (verification too frequently will affect the training speed), a validation set will be run to analyze whether theoss function and evaluation indicators are in the expected target (gradual convergence) to update the hyperparameters.

In addition, the FTN signal generation process and the adopted deepearning model are built on the basis of traditional models, which can be easily transplanted to dedicated computers (such as FPGA).

### 5.2. Simulation Results

According to the simulation results of Section 5.2.4, it can be seen that using a two-layer two-way LSTM structure for FTN signal detection will achieve relatively optimal results. The following simulation will use the above-mentioned relatively optimal LSTM structure for simulation unless otherwise specified. In addition, we convert the accuracy in the data-driven model into BER, a performance indicator in the communication field, for performance analysis.

Taking into account the feasibility of engineering implementation, we performed simulations in a variety of different multipath channel scenarios, which also proved the effectiveness of the proposed scheme to a certain extent.

#### 5.2.1. Performance Comparison between DMD-JD FTN Detection Algorithm and Pure Data-Driven Detection Algorithm

Firstly, as shown in Figure 4, shows the performance comparison of different channel equalization algorithms when the compression factor τ = 0.7 in the TU channel. The curves in different colors in the figure represent different channel equalization algorithms, followed by no equalization, ZF equalization, and MMSE equalization. Differentine shapes represent different data-driven algorithms. The redine represents the pure data-driven performance curve, which has aarge performance gap compared with the results of the DMD-JD FTN algorithm represented by the blue and black curves. The reason is that pure data driving is difficult to solve the non-linear interference introduced by FTN and the aliasing of non-linear interference through the multipath channel, which results in the mismatch of training data and test data. In Section 5.2.3, the performance of the DMD-JD FTN algorithm will also be compared with the performance of traditional model-driven algorithms.

The black and blue curves in the figure represent the performance of using MMSE equalization and ZF equalization, respectively. It can be seen that the performance of using MMSE equalization is far superior to that of ZF equalization, even reaching 3–5 dB. The possible reason is that the ZF equalization does not consider the influence of noise factors, and the results of the MMSE equalization are quite different when the noise isarge, and the performance gap is reduced when the signal-to-noise ratio is relatively high. In addition, since ZF is an unbiased estimation, the noise signal will be amplified when the signal is equalized, thereby affecting the performance of the data driving method.

#### 5.2.2. DMD-JD Algorithm Performance Using Different FTN Detection Methods in Multipath Channels

As shown in Figure 5, this is to compare the impact of different algorithms on BER performance under the TU channel compression factor τ = 0.7. It can be seen that traditional model-driven algorithms based oninear complexity have been difficult to use to achieve better results in this situation, while data-driven CNN and LSTM structures can achieve better performance than exponential complexity. Among them, the performance of the algorithm based on the Deep-Bi-LSTM structure is the best, which is about 1.6 dB higher than the performance of the MAP algorithm. The reason why such aarge gain can be obtained is that FTN detection in multipath channels can be seen as the mutual aliasing of two nonlinear processes, which will pose a greater challenge to the receiver algorithm. Traditional model-driven methods have been difficult to fit this kind of nonlinear interference. The non-linear interference introduced by FTN is a targeted detection under the condition of correct matching with a clear compression factor when the network is trained with the prior knowledge that is controllable at the transmitting end.

As shown in Figure 6 and Figure 7, the performance (blueine and blackine) of the DMD-JD detection algorithm is better than the traditional pure model-driven results in both the VA channel and the PB channel. The Deep-Bi-LSTM structure is more efficient than traditional algorithms in extracting adjacent non-linear interference from FTN. Moreover, relying on the characteristics of the network’s offline training online test number, the DMD-JD FTN detection algorithm can achieve better performance than traditional algorithms while havingower complexity.

#### 5.2.3. Adaptability of DMD-JD Algorithm to Multipath Channels



A.

DMD-JD Detection Performance Trained in the Same Channel

As shown in Figure 8, compare the generalization performance of the training model in the same channel with a fixed signal-to-noise ratio. The curves of different colors represent different multipath channels, the solidine represents the result of the normal pairing of the signal-to-noise ratio of the training data and the test data, and the dottedine represents the network trained with a fixed signal-to-noise ratio of 7 dB to test the BER performance of the corresponding channel. On the whole, the performance of using a fixed signal-to-noise ratio is somewhatost compared with the normal pairing of the signal-to-noise ratio of the training data and the test data. However, when the data-driven method is truly applied to engineering, only a fixed signal-to-noise ratio or a network model with a small number of signal-to-noise ratios need to be trained in the offline state to detect signal at all signal-to-noise ratios, which will greatly increase the reusability and practicability of the model. In addition, the simulation results show that the performanceoss in different channel conditions is acceptable, that is, the DMD-JD detection algorithm trained on the same channel has better adaptive performance. Later, what this paper aims to explain is that 7 dB is an empirical parameter, which is obtained through repeated experiments.



B.

DMD-JD Detection Performance under Different Channel Training and Signal-to-Noise Ratio Matching Conditions

As shown in Figure 9, the comparison is when the FTN signaling compression factor τ = 0.7, the training data are generated from different channel states, and the test set uses the TU channel moving speed of 30 km for testing. In the figure, different colors represent different moving rates, and differentinearities represent different multipath channels. From an overall point of view, the BER performanceoss of networks trained on different channels for TU-30KM detection is about 0.3∼0.75 dB, showing good adaptability. It can be seen that the TU channel has the worst effect when the mobile speed is 120 km. This may be due to the existence of time selective fading caused by Doppler frequency shift, and theneads to the poor matching of data set, resulting in the decline of BER performance. The performance of the PB channel when the moving speed is 3 km is relatively poor in Figure 9 and Figure 10. The possible reason is that the moving speed of the PB channel is relativelyow, and the delay spread of the channel is relatively small, which willead to the relatively insufficient feature extraction of Doppler shift and frequency selectivity for its training network. The two red curves represent the BER performance of a TU channel with a moving speed of 60 km for training, and the performance of training with a VA channel of 60 km has a small difference in the case of a higher signal-to-noise ratio. Therefore, the difference in channel state is not the only reason thateads to the generalization ability of the model. Time-selective fading and frequency-selective fading can also be used as features that affect the BER.

What needs a special explanation here is that we compared the adaptive simulation of FTN detection in multipath channels without using the channel equalization module. We found that the detection effect is very poor when the equalization module is not used, and the adaptability is almost nonexistent. The addition of equalization modules can not only greatly improve the BER performance, but also enable the FTN signaling to have a better matching relationship between the data sets in different multipath channels. The significance is that when used in actual scenarios, even if the mobile terminal is in different multipath channel switching states, it can still have a better bit error rate guarantee.

The simulation content in Figure 9, Figure 10 and Figure 11 are similar, the differenceies in the different channels represented by the test set. The performanceoss is all within 0.7 dB.

#### 5.2.4. The Impact of LSTM Network Structure on The Performance of FTN Signal Detection

From Figure 12, we can see the influence of the number of LSTMayers on FTN detection in multipath channels. Among them, the red curve represents a Single-Layer-LSTM network, the black curve represents a Double-Layer-LSTM network, and the green curve represents a Triple-Layer-LSTM network. It can be seen that for all simulated compression factors, there are cases where the performance of the Double-Layer-LSTM is better than the Single-Layer-LSTM and the Triple-Layer-LSTM. The main reason is that nonlinear interference consists of two parts. Fewerayers are difficult to extract enough features, and too manyayers can easilyead to overfitting of the trained network. In addition, an interesting phenomenon was observed in the simulation, that is, as the compression factor τ decreases, the ISI introduced by FTN becomes more and more serious, and the performance of the Triple-Layer-LSTM is getting closer and closer to the Double-Layer-LSTM. This shows that it may be more effective to use a more complex network in the case of a higher compression rate.

In addition to comparing the influence of the number of LSTMayers, the performance of traditional RNN and LSTM structures is also compared. As shown in Figure 13, the red curve represents the result using the RNN structure, and the black curve represents the result using the LSTM structure. It can be seen that under all simulated compression factors, the performance of using LSTM is better than that of RNN, and the gain range is between 0.35 dB to 1.4 dB. Compared with LSTM using the gate structure to control the weight change in the network, the weight reusability during RNN training is too high, and some slight changes will have a huge impact on the entire network so that it is difficult to obtain appropriate network parameters.

As shown in Figure 14, the influence of the Uni-LSTM and the Bi-LSTM network structure on FTN signal detection is compared. It can be seen that the overall Bi-LSTM structure represented by the blackine is about 1.8 dB better than the Uni-LSTM represented by the redine. This is because the interference of the FTN signaling comes from the previous traversal and the subsequent traversal, and the unidirectional network cannot extract the ISI from the backward propagation, which causes aarge performanceoss.

#### 5.2.5. Performance of DMD-JD Detection Algorithm in Multipath Channel under QPSK Modulation

As shown in Figure 15, the FTN detection algorithm under the condition of QPSK (4QAM) modulation and TU channel compression factor τ = 0.8 is compared. It can be seen that the DMD-JD algorithm can also be improved by about 2 dB under QPSK (4QAM) modulation.

## 6. Conclusions

This paper proposes a DMD-JD FTN detection algorithm to solve the nonlinear aliasing problem in FTN detection of multipath channels. This algorithm can not only achieve better detection performance than traditional algorithms, but also hasower complexity and stronger robustness. The transmission quality can be guaranteed in the scenario of channel switching, making the practical application of FTN possible. The core of the algorithm is that the use of data-driven methods can better fit the aliasing of nonlinear noise withower complexity, and the use of model-driven methods can enhance the robustness and generality of the algorithm on the basis of improving the performance.

The experimental results show that the DMD-JD algorithm performs well in FTN signal detection, and the performance is improved by nearly 1.6 dB compared with the traditional algorithm. Similar performance has been achieved in QPSK (4QAM) modulation, laying the foundation for future work under high-order modulation. In addition, the relatively optimal network model structure parameters of FTN signal detection are obtained through simulation and comparison. Finally, we verify that the addition of equalization modules makes the DMD-JD algorithm more robust and adaptable in multipath channel transmission scenarios.

It is worth pointing out that in order to simply verify the performance gain of the FTN detection algorithm, an ideal channel equalization method is used. In future work, we will try to use the traditional channel estimation and equalization methods from the perspectives of signaling processing and data driving to analyze the adaptability of compression factor, FTN transmission under high-order modulation, and time-frequency two-dimensional FTN signal detection. 

## Figures and Tables

**Figure 1 sensors-22-00257-f001:**
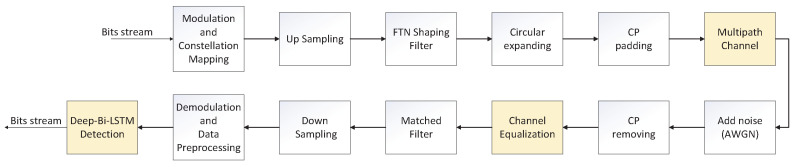
Block diagram of the transmission model of FTN signal detection communication system based on the DMD-JD algorithm.

**Figure 2 sensors-22-00257-f002:**
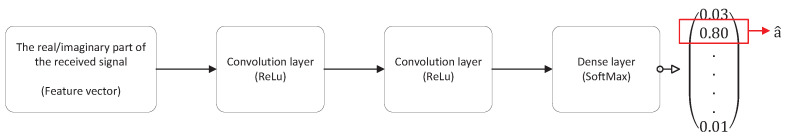
Block diagram of using CNN network structure to detect FTN signaling.

**Figure 3 sensors-22-00257-f003:**
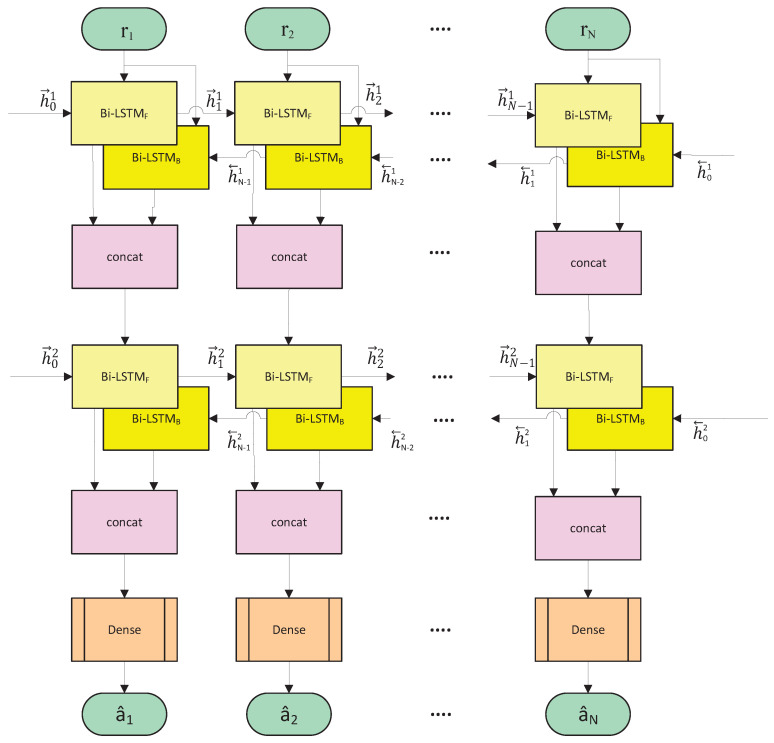
Block diagram of using CNN network structure to detect FTN signaling.

**Figure 4 sensors-22-00257-f004:**
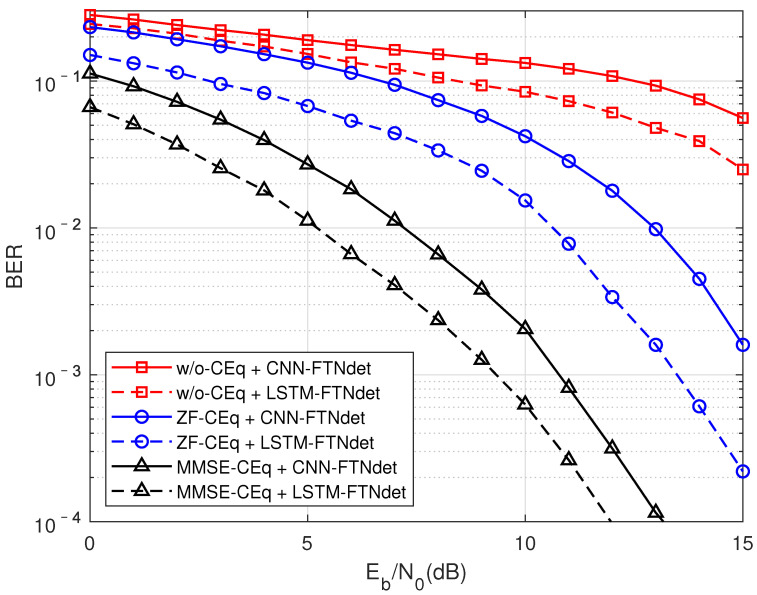
In the TU channel, when the compression factor τ = 0.7, compare the performance of different channel equalization modules.

**Figure 5 sensors-22-00257-f005:**
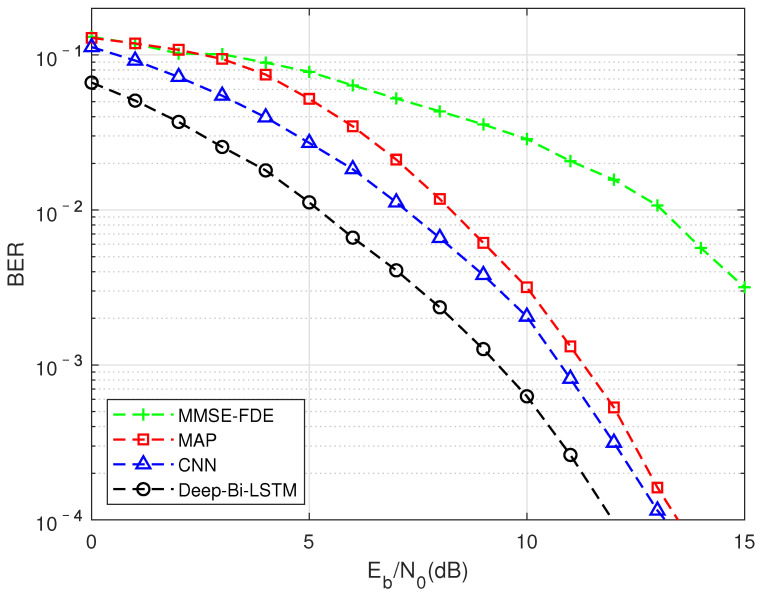
FTN detection performance results of different algorithm in the TU channel, MMSE equalization, and the compression factor τ = 0.7.

**Figure 6 sensors-22-00257-f006:**
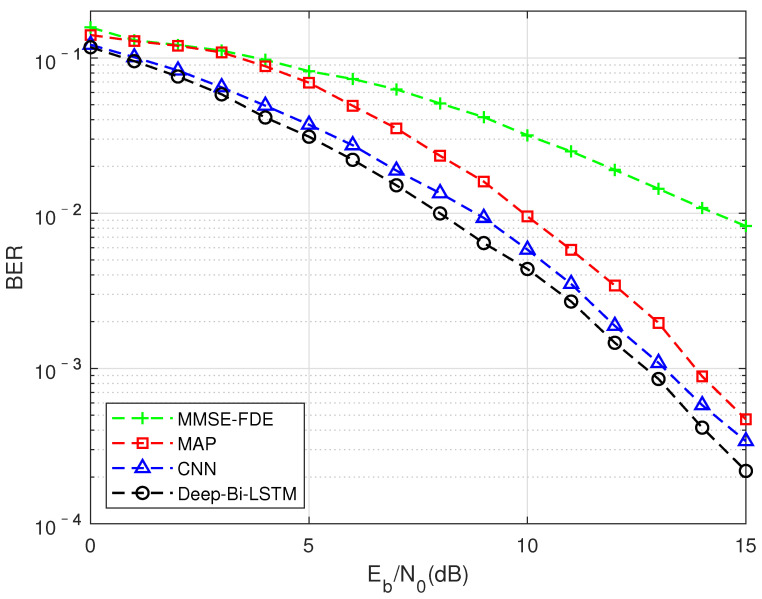
FTN detection performance results of different algorithm in the VA channel, MMSE equalization, and the compression factor τ = 0.7.

**Figure 7 sensors-22-00257-f007:**
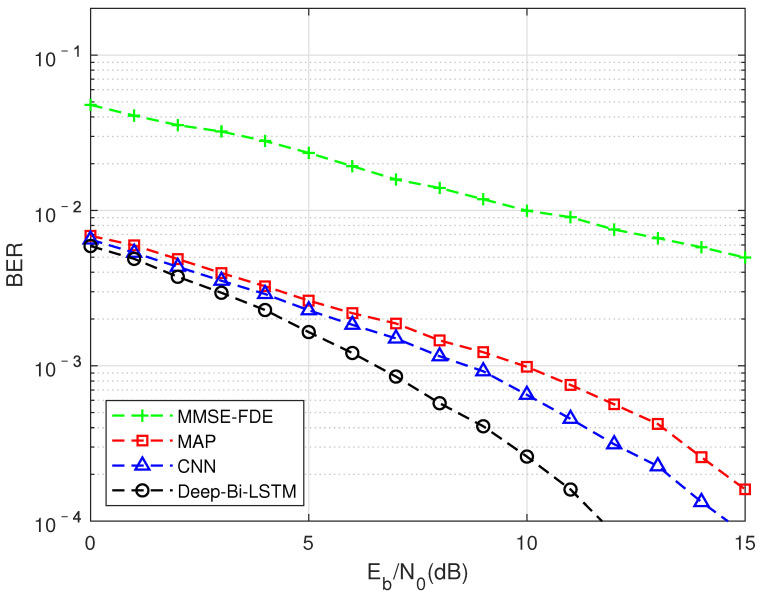
FTN detection performance results of different algorithm in the PB channel, MMSE equalization, and the compression factor τ = 0.7.

**Figure 8 sensors-22-00257-f008:**
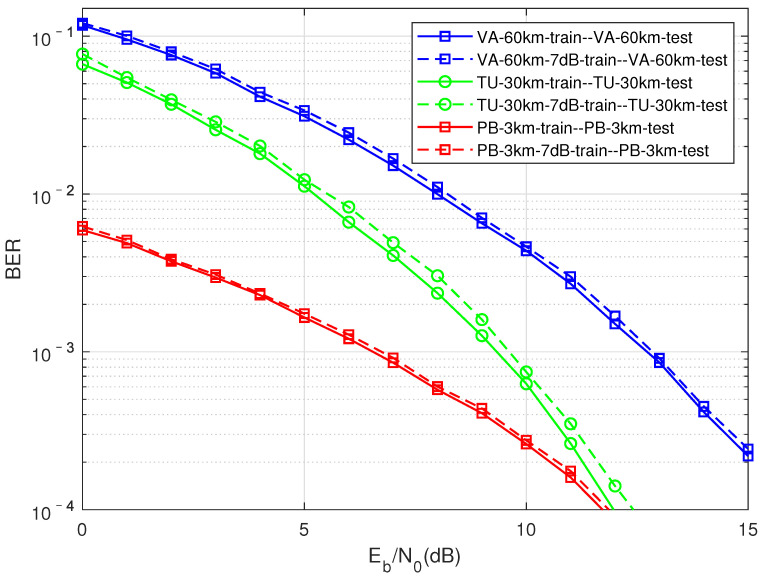
Compression factor τ = 0.7, fixed signal-to-noise ratio (7 dB) for training, and test performance for the same multipath channel with different signal-to-noise ratios.

**Figure 9 sensors-22-00257-f009:**
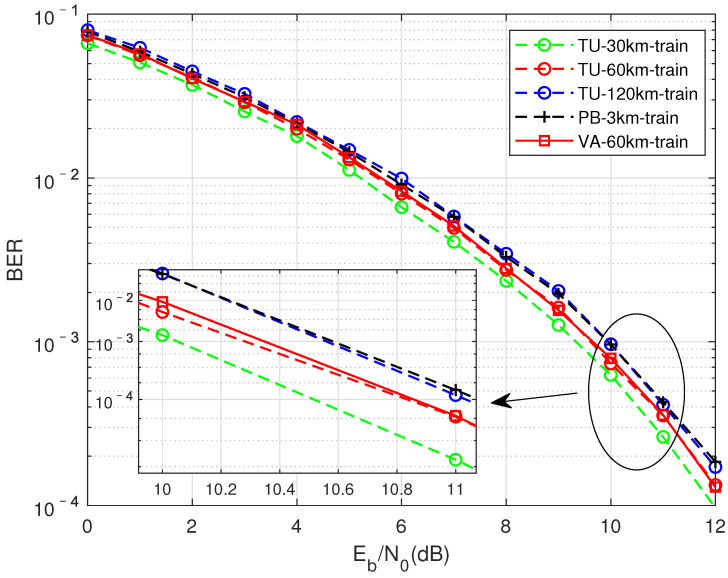
Perform training in different channel environments under the condition of matching signal-to-noise ratio, test with TU-30 km/h, performance under the condition of compression factor τ = 0.7.

**Figure 10 sensors-22-00257-f010:**
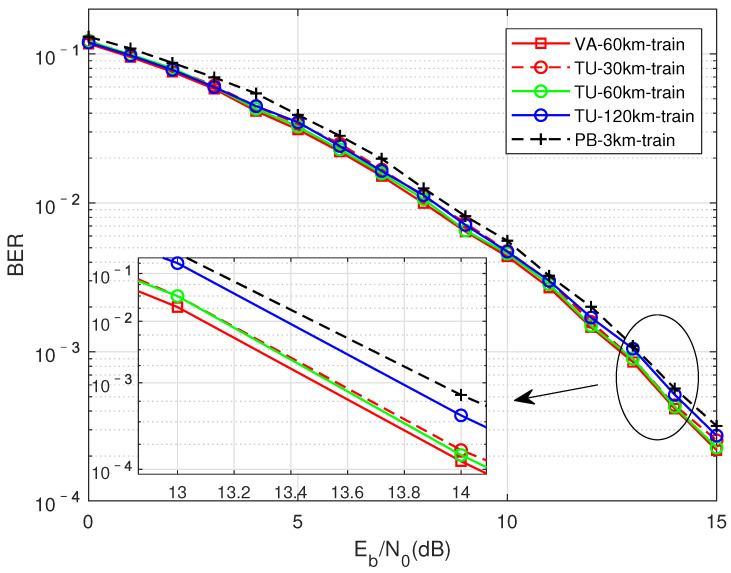
Perform training in different channel environments under the condition of matching signal-to-noise ratio, test with VA-60 km/h, performance under the condition of compression factor τ = 0.7.

**Figure 11 sensors-22-00257-f011:**
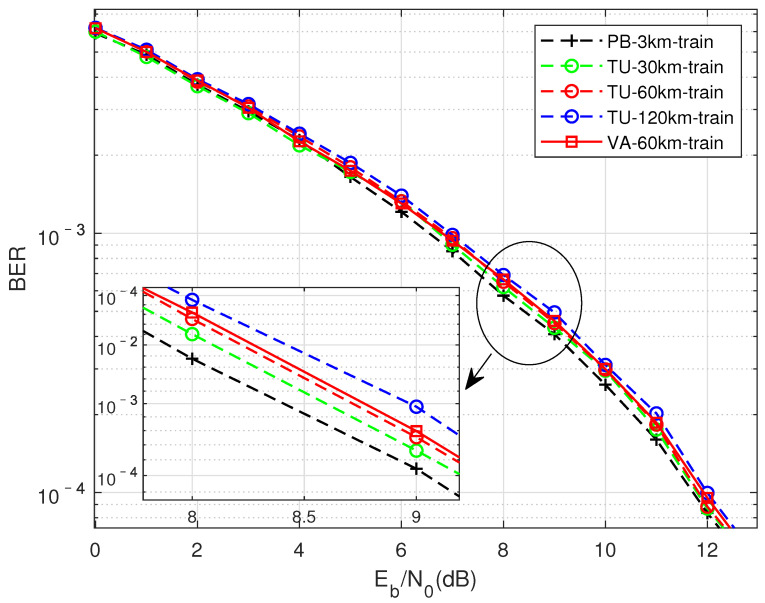
Perform training in different channel environments under the condition of matching signal-to-noise ratio, test with PB-3 km/h, performance under the condition of compression factor τ = 0.7.

**Figure 12 sensors-22-00257-f012:**
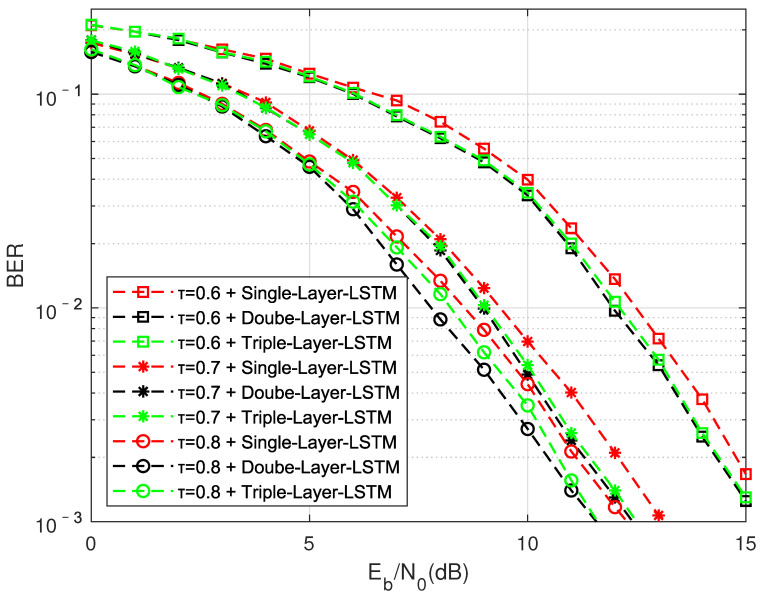
Under the TU channel moving speed of 60 km/h, compare the impact of the number of LSTM networkayers on performance.

**Figure 13 sensors-22-00257-f013:**
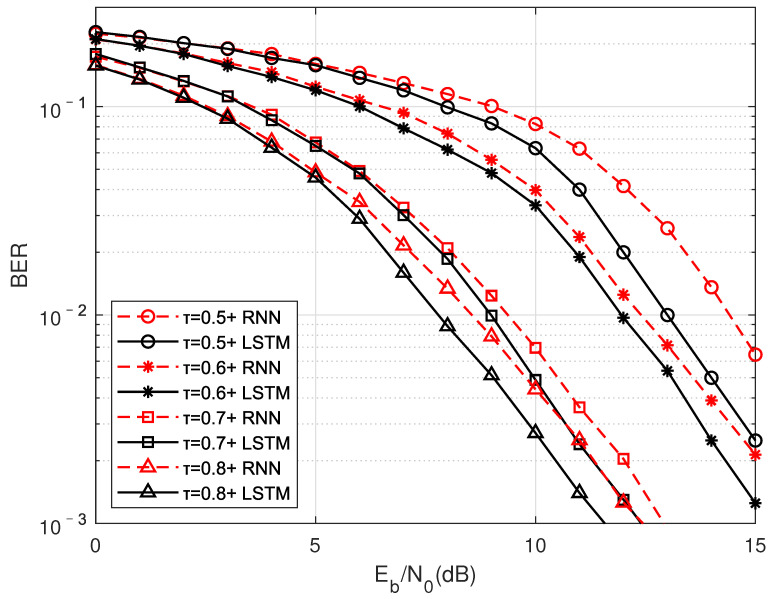
TU channel moving speed is 60 km/h, compare the performance of traditional RNN and LSTM unit.

**Figure 14 sensors-22-00257-f014:**
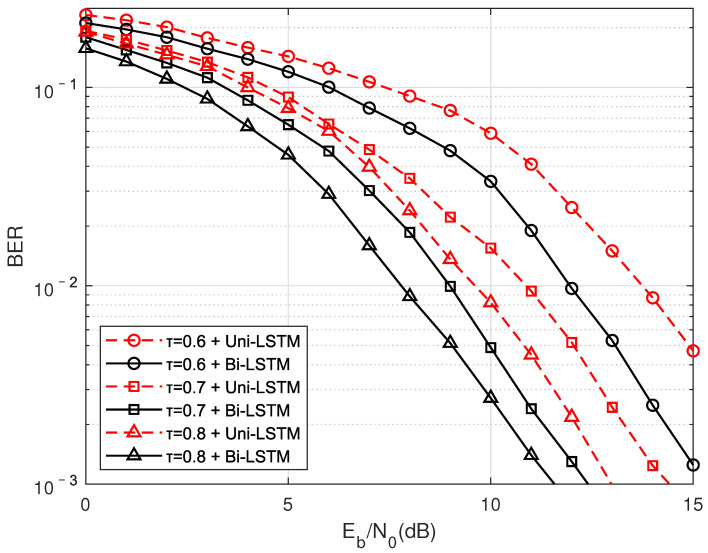
Under the TU channel moving speed of 60 km/h, compare the performance of unidirectional LSTM (Uni-LSTM) and bidirectional LSTM (Bi-LSTM).

**Figure 15 sensors-22-00257-f015:**
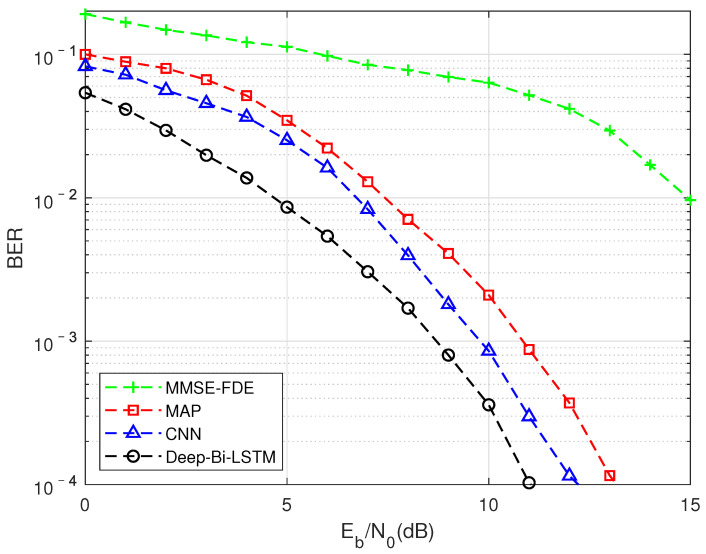
FTN detection performance of different algorithms is obtained by using QPSK (4QAM) modulation, MMSE equalization and compression factor τ = 0.8 in TU channel.

**Table 1 sensors-22-00257-t001:** Experimental settings for FTN signaling generation.

Parameter	Value
Time squeezing factor τ	0.5, 0.6, 0.7, 0.8, 0.9, 1.0
Up-sampling rate	10
Type of pulse-shaping filter	root-raised cosine (RRC)
Roll-off factor of filter	0.3
Signalength *N*	32
SNR range $E_{b}/N_{0}$	0 dB to 15 dB
Sampling frequency (MHz)	15.36
Modulation scheme	BPSK,QPSK
Channel model	AWGN, TU6, PB, VA

**Table 2 sensors-22-00257-t002:** Experimental settings in Deep Learning.

Parameter	Value
Train set size	10,000
Test set size	10,000
Numayer	6
Dropout rate	1.0
Num unit	128
Learning Rate	5 × 10${}^{-4}$
Batch Size	64
Num Epochs	10
Optimizer	Adam

**Table 3 sensors-22-00257-t003:** The delay and power of different paths in the channel model.

Channel Model	Channel Parameters
2]*TU (Typical Urban)	ζ = [ 0.0 μs, 0.2 μs, 0.5 μs, 1.6 μs, 2.3 μs, 5.0 μs ];
	P = [ −3.0 dB, 0.0 dB, −2.0 dB, −6.0 dB, −8.0 dB, −10.0 dB ];
2]* PB (Pedestrian test environment of Channel B)	ζ = [ 0.0 μs, 0.2 μs, 0.8 μs, 1.2 μs, 2.3 μs, 3.7 μs ];
	P = [ 0.0 dB, −0.9 dB, −4.9 dB, −8.0 dB, −7.8 dB, −23.9 dB ];
2]* VA (Vehicular test environment of Channel A)	ζ = [ 0.0 μs, 14.0 μs, 20.0 μs ];
	P = [ 0.0 dB, −10.0 dB, −14.0 dB ];

## Data Availability

The study did not report any data.

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
