# Peer review of "Data-Driven and Model-Driven Joint Detection Algorithm for Faster-Than-Nyquist Signaling in Multipath Channels"

_sensors, 2021, doi:10.3390/s22010257_

Round 1

Reviewer 1 Report

The paper is reasonably well written and presented. However, I have got the following issues that need to be addressed.

1. Related work: Given the wealth of research papers in the proposed field of study, it is not clear from the literature what are the research gaps are and why this study is needed in the first place. A summary table of related work might help to identify the research gap and the potential areas for contribution. 

2. Simulation results: The simulation model validation and accuracy of the simulation results have not been addressed well. The following questions need to be addressed.

What's the accuracy of the simulation results? 
How did you validate simulation models/results?

The above questions need to be discussed in Section 5.2.

Author Response

请参阅附件。

Reviewer 2 Report

The topic of the paper is relevant and is of great interest for solving the problem of FTN signaling in a multipath channel. The use of neural networks for this is also actively discussed in the literature, but in my opinion, this is not the most rational way  due to issues related to the computational complexity of the algorithms under consideration.
The work contains the section 4.4 devoted to the assessment of computational complexity, data of calculations on a general-purpose computer (5) are given, but, it is not clear to me what are the prospects for the proposed approach to be implemented on specialized computing systems: digital signal processors, FPGAs, etc., i.e. on devices that are most likely to be used for streaming signal processing.

It seems to me the work is a conceptual statement of results that cannot be reproduced or replicated.
The manuscript is clear, it is relevant for the field and it is presented in a well-structured manner.
The cited references are mostly within the last 5 years. It does not include an abnormal number of self-citations.
Figures and schemes are appropriate. They properly show the data. 
As a concept, it is a good work, which, after answering this question of the feasibility of the proposed methods (at least in general terms), can be published.

Round 2

Reviewer 1 Report

The authors have updated the paper by incorporating my comments.